

# Opinion: Strengthening Research in the Global South: Atmospheric Science Opportunities in South America and Africa

Rebecca M. Garland[1], Katye E Altieri[2], Laura Dawidowski,[3] Laura Gallardo [4], Aderiana Mbandi[5], Nestor Y Rojas[6], N'datchoh E Touré[7]

[1]Department of Geography, Geoinformatics and Meteorology, University of Pretoria, Pretoria, South Africa
[2]Department of Oceanography, University of Cape Town, Rondebosch, 7701, South Africa
[3]Comisión Nacional de Energía Atómica, Buenos Aires, Argentina
[4]Department of Geophysics, Faculty of Physical and Mathematical Sciences, & Center for Climate and Resilience Research, University of Chile, Santiago, Chile
[5]South Eastern Kenya University, Kwa Vonza, Kitui County, Kenya
[6]Department of Chemical and Environmental Engineering, Universidad Nacional de Colombia, Bogota, Colombia
[7]Université Félix Houphouët-Boigny, Abidjan, Côte d'Ivoire

*Correspondence to*: Rebecca Garland (Rebecca.garland@up.ac.za)

**Abstract.** To tackle the pressing atmospheric science issues currently and in the future, a robust scientific community is necessary in all regions across the globe. Unfortunately, this does not yet exist. There are many geographical areas that are still underrepresented in the atmospheric science community, many of which are in the Global South. There are also larger gaps in the understanding of atmospheric composition, and processes through to impacts in these regions. In this opinion, we focus on two geographical areas in the Global South to discuss some common challenges and constraints, with a focus on our strengths in atmospheric science research. It is these strengths, we believe, that highlight the critical role of Global South researchers in the future of atmospheric science research.

## 1 Introduction: Importance of reducing underrepresentation

"The future challenges for atmospheric chemistry involve nothing less than the health of the planet's climate, the health of ecosystems, and the health of humans everywhere" (National Academies of Sciences, Engineering, and Medicine, 2016). In order to address these challenges, much of atmospheric chemistry research, and atmospheric science research more broadly, works to improve the understanding of the integrated earth system and the impacts of perturbations to this system.

Such research has provided a robust evidence base to address key environmental issues, including multiple Sustainable Development Goals (SDGs). While atmospheric science research has direct links to SDGs on improving air quality (e.g., $PM_{2.5}$ levels) and climate action, there is space for further consideration of atmospheric science to support sustainable development, both in scope of issues addressed (Keywood et al., 2023), but also, as discussed here, in supporting the growth of science in underrepresented areas.



To address these large societal challenges, atmospheric science research must synthesise information not only from laboratory experiments, field measurements, satellite measurements, modelling, etc., but also information from local to regional to global scales. A further challenge is to ensure that this information is made available and understandable to the appropriate stakeholders and decision makers in an actionable form.

Despite the pressing importance of these issues, there are many geographical areas that are still underrepresented in the atmospheric science community. Furthermore, there are larger gaps in the understanding of atmospheric composition, and processes through to impacts in much of the Global South (Paton-Walsh et al., 2022; Andrade-Flores et al., 2016; Peralta, O. et al., 2023; Cazorla et al., 2022). A Web of Science search for large cities in Latin America and the Caribbean, and Africa highlights this underrepresentation as compared to well-studied cities such as London, Los Angeles and Beijing (Table 1). We believe an important part of the future of atmospheric chemistry and physics research is to work to reduce such underrepresentation.

**Table 1: Number of references from Web of science (WoS) All Databases search using Topic as city + "air quality" or city + "air pollution" (Search conducted on 25 October 2023). Population numbers are taken for cities as reported by the UN (2018) (i.e., city proper, metropolitan area, urban agglomeration).**

| City name | Number of references in WoS | Population (in thousands) 2018 | City Name | Number of references in WoS | Population (in thousands) 2018 |
|---|---|---|---|---|---|
| Abidjan | 33 | 4 921 | Beijing | 7,685 | 19 618 |
| Bogota | 223 | 10 574 | Buenos Aires | 210 | 14 967 |
| Cairo | 223 | 20 076 | Dakar | 43 | 2 978 |
| Johannesburg | 91 | 5 486 | Kinshasa | 12 | 13 171 |
| Lagos | 138 | 13 463 | Lima | 129 | 10 391 |
| London | 2,313 | 9 046 | Los Angeles | 2,715 | 12 458 |
| Luanda | 4 | 7 774 | Mexico City | 2,139 | 21 581 |
| Nairobi | 114 | 4 386 | Santiago | 622 | 6 680 |
| São Paulo | 1,499 | 21 650 | Quito | 94 | 1 822 |



It is well-documented that to understand atmospheric science, and to provide robust information to support improvements in the environment, knowledge of all regions of the globe are needed (National Academies of Sciences, Engineering, and Medicine, 2016). Thus, the research questions in these underrepresented areas are highly relevant, but still there are roadblocks

and many challenges that atmospheric researchers in these regions face (e.g. Tandon, 2021) that lead to an imbalance in atmospheric science research globally. To address these issues across the globe, a robust scientific community across the globe is necessary.

A key factor that drives much of the imbalance is funding inequities between researchers residing in these underrepresented regions compared to those in higher income countries (Fig. 1). The numbers in Fig.1 are averages for the regions; and it is

important to note that there can be a large variability within regions, but the general trends across the regions remain evident. Another key feature is that while the percentage of Gross Domestic Product (GDP) spent on research and development (R&D) is increasing at a 2-3 % compound annual growth rate in regions that include countries like China, the USA, and Europe, it is static or declining in regions that include Latin America and the Caribbean, and Sub-Saharan African countries (Fig. 1). There is also a strong relationship between research expenditure per capita and research output; for example, in the geosciences, high

income countries spent US$1064 per capita on research in 2017, while Africa spent US$42 per capita (North et al, 2020).

With the many socio-economic pressures that several of these countries in South America and Africa face, it is unlikely that this funding situation will change drastically in the near future. The ACP community should work within these known financial constraints to support, highlight, and grow research in underrepresented regions.

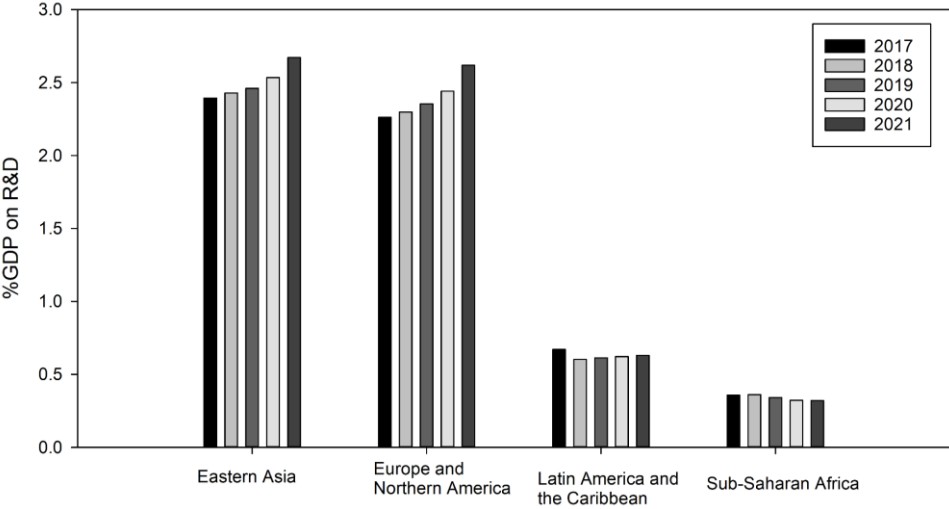

**Figure 1: Percentage of Gross Domestic Product spent on Research and Development. Data from UIS (2023a).**



Along with financial challenges, and indeed directly related to a lack of funding, is the reality that the scientific communities of researchers are smaller in these regions (Fig. 2). In Africa for example, even for a country such as South Africa, which has a well-developed scientific community compared to most African countries, the number of full-time researchers per million inhabitants averages 494 (UIS, 2023b). There are positives and negatives to the small community size, some of which are

discussed in more detail below. The reality is that a lack of financial and human capital influence the amount and type of research that can be conducted in these regions.

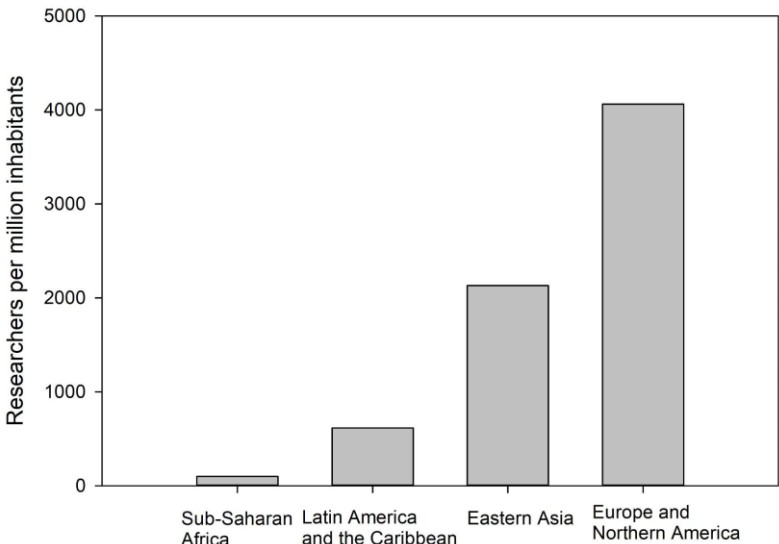

**Figure 2: Researchers (in full-time equivalent) per million inhabitants across regions in 2020. data from UIS (2023b).**

In this opinion piece, we focus on two geographical areas, Africa and South America, that we found have many similarities in the challenges and constraints, as well as strengths in atmospheric science research. It is these strengths, we believe, that highlight the critical role of Global South researchers in the future of atmospheric science research. We acknowledge that circumstances vary widely across these regions, as well as across the Global South, and thus we cannot capture all researchers' experiences in this piece. But rather, these are our reflections as researchers living and working in these regions.

While other reports have focused on the unique research questions in our areas (e.g., Burger et al., 2023; Cazorla et al., 2022; Huneeus et al., 2020a; Molina et al., 2015) here we focus on the strengths of our approach to research. From our experience, the data scarcity and smaller budgets of our regions have bred creative approaches to atmospheric science research that are



highly relevant to the future of the field. Of course, there remain stumbling blocks that we have identified, and these are issues that the ACP community can work together to address.

## 2 Innovative approaches when data are scarce

In general, the two regions of discussion are data scarce with respect to measurements of relevance to atmospheric chemistry. However, within South America and Sub-Saharan Africa there is a gradient from some relatively well-studied regions (e.g., Johannesburg-Pretoria mega city in South Africa (e.g. Borduas-Dedekind, 2023; Lourens et al., 2012; Hersey et al., 2015) and cities like Mexico City, Mexico, São Paulo, Brazil, and Santiago, Chile (Andrade-Flores et al., 2016) to areas with a complete paucity of data (discussed in Paton-Walsh, et al., 2022). There is also variability in the pollutants that are measured. For example, the map of data on OpenAQ (https://explore.openaq.org/) highlights that in these regions, while PM monitoring is still scarce, there are even fewer trace gas measurements (e.g., ozone, VOCs, and $NO_2$). In addition, PM monitoring focuses on mass concentrations, with more detailed analyses (e.g., size distributions, composition) are often lacking.

Given the lack of funding and the small community size, it is not possible to always use approaches that are state-of-the-art, cover large areas, or that require large amounts of human capital. However, a scarcity of resources can drive the development of new and innovative approaches to answer atmospheric science research questions, as well as efficient use of resources. The measurement might be a standard one, but the approach needed to conduct the measurement and the place-specific findings and implications can be novel. Examples of studies in unique Global South locations include the study of ozone at a high-altitude and pristine station in Chile that due to the dominance of clean Pacific Ocean air masses was able to record the influence of El Niño and La Niña episodes on background ozone (Anet et al., 2022), a study in coastal South Africa focused on aerosol size distributions that was able to observe sea spray generation with almost unlimited fetch from the Southern Ocean (February et al., 2021), as well as studies examining the interactions between large natural and anthropogenic sources, such as in the Amazon Basin (Nascimento et al., 2021).

A scarcity of resources can also result in the use of fairly basic approaches, or those that are seen as explicitly not novel as they are well-used and well-known approaches in the Global North. It raises a question about how we define novelty as a community. If a measurement has been made 1000 times in the Global North, but never in an African country, then we argue that it is indeed a novel measurement and important for improving our understanding of atmospheric chemistry processes and variability. When publishing analyses from these regions, the response we often receive is that the findings are "too local" or "not of interest to the global community" and/or deemed not to be novel. This is described in a recent article on the impact of Global South research (Wild, 2023), and a quote within, "When researchers in the global North produce research, it's understood as if it was universal, whereas when research is done in the global South, then it's only local and applicable to those settings."



These are important issues to keep in mind when reviewing papers from these regions. An attempt should be made to evaluate the novelty of the research in the context of the financial and human resources available, as well as the available background

scientific knowledge of atmospheric science in the region. Research in well-studied areas can focus on highly specific and detailed questions due to the history of contextual research at the site or in the region that leads to a large amount of background knowledge. Those older and contextual studies were considered novel at the time they were published. Logically then, basic contextual studies in data scarce regions of Sub-Saharan Africa and South America should also be considered novel as they will set the stage for future process-level and "novel" research.

**3 Transdisciplinarity and integrative approaches**

Another approach to addressing issues of resource scarcity and data paucity is to take a broader and more integrative approach to conducting research. In some cases, this might mean combining a number of different information sources to overcome the uncertainty of any one. For example, remote sensing and satellite data frequently lack local validation, and local measurements are sometimes low in frequency due to power outages, maintenance, spare parts or staffing issues, but used in conjunction they

can address research questions that they could not alone.

In some instances, a push towards inter- and transdisciplinary research can result from multiple domains coming together to address research questions of global and local relevance. An example of this is one of the South African Research Infrastructure Roadmaps, BIOGRIP. The Biogeochemistry Research Infrastructure Platform (BIOGRIP) is a South African research initiative that drives discovery in how biological, geological, chemical and physical processes interact to shape natural

environments over time and space (www.biogrip.ac.za). This initiative has expanded the definition of biogeochemistry in South Africa to include understanding how Earth systems interact from early Earth history to the present and through to the future. This includes research questions around the origins and diversification of life, which can only be addressed in Sub-Saharan Africa where humans originated, through to the impact of human activity on the environment, which is a truly global issue.

Another example comes from West Africa, where the West African Science Service Centre on Climate Change and Adapted Land Use (WASCAL), has contributed to large-scale climate-focused research for more than a decade. With the support of the German federal Ministry of Education and Research (BMBF), WASCAL works to train young scientists in diverse climate change and atmospheric chemistry topics through its postgraduate and doctorate schools across the region. WASCAL also provides information and knowledge across several scales (local, national, and regional levels) to its West African member

countries to help cope with the adverse impacts of climate change and to devise integrated mid and long-term strategies to build up resilient and productive socio-ecological landscapes.





In South America, we find integrative studies as well. For instance, Huneeus et al. (2020b) addressed particle pollution in central and southern Chile focusing not only on air quality and atmospheric circulation, but also considering the underlying socio-economic drivers of wood burning in the region characterised by energy poverty.

## 4 Research to support action


Atmospheric science has a strong history of providing a robust evidence base to support policies to address key environmental issues such as stratospheric ozone depletion, acid deposition, and climate change. This characteristic of atmospheric science will continue to be important for the field into the future; this is an area where we find many researchers in resource-constrained areas excel. The cumulative experience of researchers from the Global South regarding the science-policy making interface
may be of interest for researchers in the Global North.

As funding for research is small and there are other urgent pressing issues that countries face, researchers have to articulate and show societal impact much more explicitly. This also encourages researchers to foster relationships with local stakeholders, including policymakers. These are small and interlinked communities and then interactions with policymakers are integrated into projects from the beginning.

This is a feature of the way researchers in the Global South use transdisciplinarity and the layering of different fields of research to address complex problems of local and global impact. Another feature common to these regions in the Global South is that due to the small community size, most scientific experts who work in policy-relevant fields work with one another and with policymakers. This results in more regular exchange of ideas and increases the potential impact of research on policy.

An example of this on a larger scale is the recently released Integrated Assessment of Air Pollution and Climate Change for
Sustainable Development in Africa (UNEP, 2023) which brought together over 100 African-based researchers as authors of the report who also worked to develop and analyse the impacts of future emission scenarios over Africa. The assessment process also included policymakers and regional organisations across Africa who provided input and review of the process and the report. The models and findings from the Assessment can now be used by local researchers and policymakers working together to interpret and downscale the results for their local contexts (Kaudia et al., 2022).  Such an assessment was previously
performed for the Latin America and Caribbean (UNEP and CCAC, 2016); both of these provide science-based policy analyses of scenarios to decrease emissions of SLCPs, air pollutants and GHGs.

Due mostly to data scarcity, our research findings can often have larger uncertainties than those in well-resourced countries. However, policy decisions must be made within these uncertainties. Thus, we also have to be adept at communicating uncertainties and their implications to a wide-range of stakeholders. For example, Gallardo, et al. (2018) describes the complex
process and considerations for providing an evidence base for policy (focusing on air quality and mobility) with such



uncertainties in Santiago, Chile. While uncertainties and gaps in knowledge exist in Santiago (for example in emission inventories and comprehensive ambient monitoring of pollutants) science has effectively supported policy in the past and recommendations for the future of the transport sector are detailed based upon the available scientific evidence.

Communicating these uncertainties and their implications is a key responsibility for researchers who work to support policies; with limited budgets, it is imperative that effective solutions are prioritised. While this is true in all countries, in such resource-constrained contexts, the opportunity cost of ineffective policies is more consequential. This is a key point for why "helicopter science" (as defined below) can be so problematic.

## 5 Equitable collaborations are needed to support the growth of research communities

For the research community as a whole, collaborative research is critical to address complex atmospheric science research questions as well as impacts. This includes collaborations across disciplines and across borders. However, when inequalities are present in collaborations (e.g., differences in funding, access to equipment, number of researchers), the collaborations that form can also be inequitable. The inequities in these collaborations diminish the potential of the impact on science and policy. Indeed, the linkages to the local communities are what is needed to ask relevant research questions, analyse outputs in the local context, and provide outcomes that align to needs. Misalignment can lead to decisions made on incomplete or incorrect information, which, as discussed above, can have large negative consequences.

"The lack of capacity" is often highlighted as a roadblock for research communities in underrepresented regions. Such framing leads to many "capacity building" efforts that ignore the local capacity and expertise, and oftentimes small and short-term funds are then directed only at these efforts to the exclusion of other equally pressing research priorities. Constraints in capacity are rooted in part in smaller communities (Fig. 2). Thus, while our communities do have deep expertise in specific aspects of atmospheric science, they are generally much smaller communities and thus there are areas of expertise that are still missing. Equitable collaborations with the global community can augment the expertise of local communities to explore complex atmospheric science problems together. Regional networks of atmospheric scientists, such as those under IGAC (i.e., Americas working group for Latin America and the Caribbean (https://igacproject.org/working-groups/AmericasWG) and ANGA for Africa (https://igacproject.org/working-groups/anga)) are a key resource in linking local and international researchers to support equitable collaborations.

To be equitable and effective, the nature of the collaboration, especially in cases where there may be an imbalance in resources, is key and needs to be thoughtfully and deliberately planned and executed (see Text Box 1). The SAFARI2000 campaign is an example of a project that aimed towards a transformational collaboration, articulated this vision in Annegarn and Swap (2012), and, it can be strongly argued, that it achieved this aim. The inclusion of local researchers as leaders in large international projects is a key characteristic of more equitable collaborations as can be seen in some projects in these regions



such as, AMMA (Lebel et al., 2011); DACCIWA (Knippertz et al., 2015; Evans et al., 2018); GAPS-megacities (Saini et al., 2020); MILAGRO (Molina et al, 2010); PAPILA (Castesana et al, 2022).

Equitable collaborations have both short-term positive impacts (e.g., in specific projects), but also can have long-term positive impacts in helping to support the growth of a flourishing local research community. More so than a once-off capacity building
workshop. Without such a community, capacity building efforts will have little impact as there won't be a community for such trained researchers to join. To address atmospheric science issues across the globe, a robust scientific community across the globe is necessary.

---

**Text Box 1: Summary of types of scientific relationships**

Scientific relationships can be described as different types including exploitative, transactional or transformational (Annegarn and Swap, 2012, Clayton et al., 2010).

A type of exploitative collaborations is "parachute science" or "helicopter science", where scientists from generally more "...well-resourced countries/settings perform research in resource-poor settings with limited to no involvement of local communities or researchers." (EGU, 2023)

Transactional relationships may have some involvement of local researchers and stakeholders, but "...little effort is made to engage the local scientists as intellectual partners or to nurture local students and institutions" (Annegarn and Swap, 2012).

Transformational relationships should be the goal for equitable collaborations. In these relationships, the process from proposal to project is open and mutually beneficial, where responsibilities and intellectual leadership is shared.

---

## 6 Concluding thoughts

When considering air pollution, air inequality exists within and between countries, where the poorest, most vulnerable, and those who contribute the least are exposed to the highest levels of air pollution. Many of the areas within the regions discussed here have high, and sometimes growing, levels of air pollution, yet are less equipped to invest in the science to understand and improve the situation (Fig. 1 and Fig. 2). This inequality has detrimental impacts on these regions. However, it also has global impacts as, in order to tackle the pressing atmospheric science issues currently and in the future, a robust scientific community
across the globe is necessary. This does not yet exist due to many challenges, some of which are highlighted here. Action is needed across the broader research community including funders, publishers, and researchers to address these challenges. First-rate atmospheric science communities with strengths that align directly with the future needs of atmospheric science have developed and are growing in our regions despite these challenges. With a focus on equitable collaborations and



transformational relationships, the atmospheric science community can work together to continue to increase capacity and
address complex research challenges that are critical for human and ecosystem health as well as climate.

**Competing interests**

RMG, KEA, NET are members of the editorial board of Atmospheric Chemistry and Physics.

**Acknowledgements**

The authors appreciate earlier discussions with the IGAC SSC members on issues facing researchers in the Global South.

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
