# Peer review of "Opinion: Strengthening Research in the Global South: Atmospheric Science Opportunities in South America and Africa"

_EGUsphere, 2023_

## Referee Comment (RC2)

Opinion: Strengthening Research in the Global South: Atmospheric Science Opportunities in South America and Africa

**Manuscript Number: 2023-2566**

**Summary**
The authors presented Africa and South America as the two geographical areas in the Global South that are underrepresented in the atmospheric science community. The article identified some common challenges and constraints hindering the development of atmospheric science research in the regions. They also highlighted the strengths of the researchers in the regions and their   critical role in the future of atmospheric science research

**Technical observations**

Line 38 – 39: References should be cited in chronological order according to the Journal style

Line 88: References should be cited in chronological order according to the Journal style

**Recommendation**

The article is indeed an opinion which reflected the true situation in the research community in the regions of interest. The article could add value to the scientific community of published. I recommend it for publication after appropriate editorial work

---

## Author Comment (AC1)

Opinion: Strengthening Research in the Global South: Atmospheric Science Opportunities in South America and Africa

Thank you to the reviewers for the helpful comments. We have addressed the comments as detailed below. The reviewers' comments are in italics and our responses are in blue

**Reviewer 1:**

*As a global south researcher, I think this article touches on some critical points within the broader atmospheric science community. My comment for minor revision stems from the author's title, "Strengthening Research in the Global South: **Atmospheric Science** Opportunities in South America and Africa". For me, there is a strong argument that this should include some examples of collaboration within climatology and meteorology, instead, there is a significant focus on chemistry (justifiably so within the context of the journal).*

We have updated the text to be more inclusive of atmospheric science. This has included adding examples that are outside of atmospheric chemistry for both regions. The changes are in track changes in the updated opinion piece.

*A second valuable analysis would be to see the number of papers from the WoS results (table 1) that:*

*Was, first authored by a "local" researchers/institution,*

*was written by a team compromised fully of local researchers vs international collaboration and,*

*funded by an international partner vs a local partner.*

*This could subsequently be visualised in a simple graph, which has a visual impact.*

We have updated the table to a graph to better visualize the point that the number of papers for large cities in South America and Africa have far fewer papers on air pollution than example cities in the US, Europe, and China. In this graph, we are not trying to discuss helicopter science, but rather the underrepresentation in the field.

The points that the reviewer highlights would be helpful to understand the level of helicopter science. Using WoS, it was possible to assess the percentage of papers that had an author with an affiliation from a local institution (i.e. same country as the city). It was not possible to easily assess the other aspects recommended by the reviewer.

This draft figure is below. As can be seen, the results are not clear that there are large differences between these cities and there is a lot of variability. Thus, more analysis would be needed to better understand these trends in helicopter science. This would be interesting to explore; however, as the figure in the opinion is focusing more on underrepresentation than helicopter science, we have not included the below figure nor the additional needed analysis in the opinion piece.

[Figure]

*Other than these minor revisions, I believe the paper is valuable and insightful, and this conversation should continue in the community.*

Thank you.

**Reviewer 2:**

**Summary**

*The authors presented Africa and South America as the two geographical areas in the Global South that are underrepresented in the atmospheric science community. The article identified some common challenges and constraints hindering the development of atmospheric science research in the regions. They also highlighted the strengths of the researchers in the regions and their   critical role in the future of atmospheric science research*

***Technical observations***

*Line 38 – 39: References should be cited in chronological order according to the Journal style*

*Line 88: References should be cited in chronological order according to the Journal style*

The in-line references have been updated.

***Recommendation***

*The article is indeed an opinion which reflected the true situation in the research community in the regions of interest. The article could add value to the scientific community of published. I recommend it for publication after appropriate editorial work*

Thanks.